# Neuralgic Amyotrophy with Concomitant Hereditary Neuropathy with Liability to Pressure Palsy as a Cause of Dropped Shoulder in a Child after Human Papillomavirus Vaccination: A Case Report

**DOI:** 10.3390/children9040528

**Published:** 2022-04-07

**Authors:** Hye-Chan Ahn, Do-Hoon Kim, Chul-Hyun Cho, Jun-Chul Byun, Jang-Hyuk Cho

**Affiliations:** 1Department of Rehabilitation Medicine, Keimyung University Dongsan Hospital, Keimyung University School of Medicine, Daegu 42601, Korea; plokij36@naver.com; 2Department of Laboratory Medicine, Keimyung University Dongsan Hospital, Keimyung University School of Medicine, Daegu 42601, Korea; kdh@dsmc.or.kr; 3Department of Orthopedic Surgery, Keimyung University Dongsan Hospital, Keimyung University School of Medicine, Daegu 42601, Korea; oscho5362@dsmc.or.kr; 4Department of Pediatrics, Keimyung University Dongsan Hospital, Keimyung University School of Medicine, Daegu 42601, Korea; goodpeddr@naver.com

**Keywords:** brachial plexus neuritis, hereditary sensory and motor neuropathy, paralysis, vaccination, pediatrics

## Abstract

Hereditary neuropathy with liability to pressure palsy (HNPP) makes nerves increasingly susceptible to mechanical pressure at entrapment sites. Neuralgic amyotrophy (NA) can cause sudden regional weakness following events to which the patient is immunologically predisposed, such as vaccination. However, NA related to human papilloma virus (HPV) vaccination is seldom reported. We describe the case of a child with NA as the cause of a dropped shoulder following the administration of the HPV vaccine. Underlying asymptomatic HNPP was confirmed in this patient based on the electrodiagnostic findings and genetic analysis. We speculate that HPV vaccination elicited an immune-mediated inflammatory response, resulting in NA. Our patient with pre-existing HNPP might be vulnerable to the occurrence of an immune-mediated NA, which caused the dropped shoulder.

## 1. Introduction

Hereditary neuropathy with liability to pressure palsy (HNPP) is a rare autosomal dominant peripheral nerve disorder [1]. Clinical features include painless, recurrent, and transient weakness at entrapment sites or susceptible pressure points [2]. The frequently involved patterns are similar to those seen in entrapment neuropathy; however, brachial plexus involvement is uncommon [1,2].

Neuralgic amyotrophy (NA) is a markedly underdiagnosed or misdiagnosed peripheral nerve disease due to the heterogeneity of clinical appearance [3]. It represents a sudden onset of paralysis, atrophy, and sensory deficits in the shoulder region with a preceding episode of severe pain [4]. Although the exact pathophysiology of NA has not yet been established, it is presumably associated with inflammatory autoimmune pathophysiology [3,4]. Vaccination is less commonly known to be related to the occurrence of NA [4,5].

This study was approved by our institutional review board (IRB number: 2021-01-079). Written informed consent was obtained from the patient’s legal guardian. This report describes the case of a female child who presented with a dropped shoulder due to NA following administration of the human papillomavirus (HPV) vaccine. We found that she also had underlying HNPP during the diagnostic process. There are currently no published cases of NA with concomitant HNPP occurring after HPV vaccination. HPV vaccination is suspected to elicit an immune-mediated inflammatory response, resulting in NA. Patients with pre-existing HNPP, although asymptomatic, might have potential vulnerability to the occurrence of an immune-mediated NA, thus causing symptoms, such as dropped shoulder.

## 2. Case Report

A 12-year-old female child received her first dose of Cervarix (GSK, Brentford, UK) vaccination for HPV prophylaxis in the left deltoid muscle. Five days after vaccination, the child complained of a sudden onset of painless dropped shoulder on the left side. She denied any history of trauma, notable physical exercise, or previous medical illness. No adverse events had occurred following the suggested pediatric immunization program and she had not been vaccinated for coronavirus disease 2019 (COVID-19). The patient’s parents denied any family history of notable genetic diseases. On initial physical examination, there was Medical Research Council (MRC) grade 2 weakness of the shoulder abductor and flexor muscles and grade 3 weakness of the internal and external rotator muscles without atrophy. The sensation of the left upper lateral limb was altered on light touch. Deep tendon reflexes were normal, and no upper motor neuron lesion signs were observed.

Laboratory findings—including C-reactive protein, erythrocyte sedimentation rate, creatine kinase, antinuclear antibody, rheumatoid factor, and anti-GM1 ganglioside antibodies—were negative. We have also excluded infectious neuritis, including human immunodeficiency virus infection and Lyme disease. Magnetic resonance imaging (MRI) of the brain, cervical spine, and brachial plexus performed one week after symptom onset revealed no abnormal findings. Nerve conduction studies revealed the absence of the left lateral antebrachial cutaneous sensory nerve action potential (SNAP) and decreased amplitude in left axillary compound motor action potential (CMAP). It also showed generalized demyelinating polyneuropathy, including the following signs: slowing of the bilateral median, bilateral ulnar, bilateral common peroneal, and bilateral tibial nerve conduction velocity; prolonged distal latency of bilateral median SNAP; and borderline distal latency of bilateral median, bilateral ulnar, and bilateral common peroneal CMAP close to the upper limit of normal (Table 1). Needle electromyography showed active denervation potentials of the left supraspinatus, deltoid, and brachioradialis muscles. It also showed decreased motor unit recruitment in the left shoulder girdle muscles. This electrodiagnostic evaluation revealed left brachial plexopathy predominantly involving the upper trunk with generalized demyelinating polyneuropathy.

Based on her medical history and the results of our evaluations, the patient was diagnosed with NA with peripheral polyneuropathy. Cerebrospinal fluid analysis to rule out acute or chronic inflammatory demyelinating polyneuropathy showed no abnormal findings. Genetic analysis was performed to identify the cause of hereditary polyneuropathy. Multiple ligation probe amplification (MLPA) was performed by probe mixes P033-B4 to detect copy number variations (CNVs) in the *PMP22* gene (MRC-Holland, Amsterdam, The Netherlands). Coffalyser.Net (MRC-Holland) was used for fragment analysis. The height ratio of the polymerase chain reaction (PCR)-derived fluorescence peaks was measured to quantify the amount of PCR products after normalization, and CNVs were identified when the ratio was <0.65 or >1.3. The MLPA revealed a contiguous heterozygous gene deletion of chromosome 17p11.2 that includes *COX10*, *PMP22*, and *TEKT3*, which confirmed the diagnosis of HNPP (Figure 1).

The patient received oral prednisolone for 10 days (40 mg/day for 5 days, the dose of which was then tapered for another 5 days) and underwent two courses of this treatment. Five months after the onset of symptoms, the patient recovered completely.

## 3. Discussion

HNPP is a hereditary neuropathy caused by deletion of the *PMP 22* gene that results in sausage-like focal thickening of the myelin sheath [1]. It is mostly diagnosed in early adulthood between 20 and 30 years of age, or if there is a family history [2]. HNPP is often triggered by physical activity, trivial compression, and negligible trauma that affects transient and recurrent focal neuropathy [2].

NA typically represents a preceding episode of severe pain before the sudden onset of paralysis [5]. However, some patients affected with NA complain of painless weakness in the region, such as in our case [4,5,6]. NA usually occurs between the age of 20 and 60 years. In addition, there is a biphasic peak of incidence in pediatric cases during the neonatal period and adolescence [4]. More than 50% of NA cases have a trigger event such as infection, vaccination, surgery, pregnancy, trauma, or stress, which activates the immune-mediated inflammatory response in these patients [6]. Vaccination is considered a rare trigger, with approximately 4.3% of antecedent events of NA [4,6]. NA also occurs as a hereditary disease, which is an autosomal dominant disorder with characteristic features such as earlier onset, higher recurrence rate, and worse long-term prognosis when compared with the idiopathic type. In 55% of the affected families, NA is associated with a point mutation or duplication in the SEPT9 gene on chromosome 17q25.3 [4,6,7]. In our case, the patient experienced her first shoulder drop following an HPV vaccination, and if NA recurs, genetic testing of the SEPT9 gene must be considered to discriminate hereditary NA.

HPV infection—caused by HPV types 16, 18, 31, and 45—is the major cause of cervical cancer [8]. HPV vaccination provides prophylaxis against HPV infection and related diseases, such as genital warts and cervical cancer [9]. Furthermore, even in previously HPV-infected women, HPV vaccination is important because it helps prevent the spread of infection to others [10]. This vaccination is advantageous in conferring protection against persistent HPV type 16 and 18 infections for 7 years. Moreover, HPV type 16 and 18 infections are more likely to be eliminated in vaccinated women than in unvaccinated women [11]. It appears to affect an earlier clearance of HPV infection in patients who have tested positive for HPV DNA [10].

The common adverse effects of an HPV vaccine are injection site discomfort—such as pain, swelling, and redness—due to an inflammatory response to virus-like particles [8,9]. It can also lead to systemic symptoms such as fever, nausea, vomiting, headache, dizziness, myalgia, and diarrhea [9]. Cases of autoimmune diseases following exposure to HPV vaccination are reported regularly, although there is no strong evidence that HPV vaccination increases the risk of autoimmune disease [12,13]. Thus, it would be necessary to verify more cases and constantly perform surveillance. In previous studies, only two cases of adverse effects presenting as NA after HPV vaccination have been reported to date, in which the patients presented with severe pain and paralysis in the region after the second HPV vaccination [14,15]. In our case, the patient complained of painless weakness in her shoulder after the first injection. We presumed that our patient presented with symptoms even after the first HPV vaccination because of the underlying HNPP. It might have been due to hereditary vulnerability to the occurrence of an immune-mediated NA.

Our patient only had a clinical presentation of NA after HPV vaccination; however, the electrodiagnostic findings led us to suspect underlying diffuse polyneuropathy. Inflammatory demyelinating polyneuropathy was excluded because of the negative results of cerebrospinal fluid analysis and brachial plexus MRI. Genetic testing demonstrated *PMP 22* deletion, which is the key feature of HNPP. The child did not meet the other criteria for HNPP diagnosis, which are as follows: autosomal dominant family history, age at symptom onset around the second decade, and rapid onset sensorimotor deficit preceded by a minor injury [1]. However, HNPP in pediatric patients has been reported as a phenotype of diffuse demyelinating polyneuropathy [2]. The parents of our patient declined the offer of genetic counselling; thus, we could not obtain information regarding family history of HNPP, which was a limitation of this case study.

In our patient, NA was potentially triggered by an immune-mediated inflammatory response due to HPV vaccination [5]. Remiche et al. reported that genetically determined neuropathy might affect immune-related peripheral neuropathy [16]. It has been suggested that hereditary neuropathy could predispose patients to the development of immune-mediated neuropathy [16]. In previous studies, cases of the associations between genetic and inflammatory neuropathies have been reported; thus, it might be necessary to consider genetic disorders in the process of diagnosing neuropathy [17]. The patient’s concomitant HNPP vulnerability to pressure may be severely worsened due to virus-like particle-related immune reactions against the brachial plexus near the vaccination site.

To the best of our knowledge, this is the first case reporting the comorbidity of NA after HPV vaccination and underlying HNPP. HPV vaccination may potentially cause NA in patients with pre-existing HNPP due to an immune reaction. The asymptomatic HNPP might be vulnerable to the occurrence of an NA as the cause of a dropped shoulder. Our case report demonstrates a possible association between the NA and HNPP. The observation of potential factors that might suggest HNPP in patients with NA by comparing the differences between the reported cases will be of great importance and worth examining in future studies. This study highlights the importance of assessing clinical, electrodiagnostic, and genetic findings to make an accurate diagnosis of NA after HPV vaccination in a patient with pre-existing HNPP. Clinicians should consider hereditary neuropathy as a possible predisposing disease for the development of immune-mediated neuropathy. There may be genetic and immunological associations between HNPP and NA, which require further investigation.

## Figures and Tables

**Figure 1 children-09-00528-f001:**
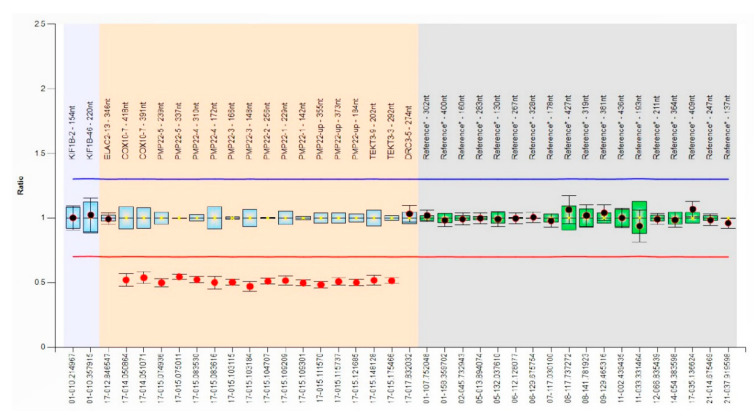
MLPA results of samples with contiguous gene deletion of chromosome 17p11.2 that included *COX10*, *PMP22*, and *TEKT3* in the bar chart generated using Coffalyser.Net. The x- and y-axes indicate the names of each MLPA probe and the peak ratio, respectively. The reference probe area is colored in gray. Deletions or duplications can be identified when the peak ratio was <0.65 (red line) or >1.3 (blue line), respectively. Exons with a reduced peak ratio (heterozygous deletion, between 0.40 and 0.65) are indicated with a red dot.

**Table 1 children-09-00528-t001:** Nerve conduction study results at presentation.

Nerve	Recording Site	Latency (ms) [Normal Value]	Amplitude (µV and mV) [Normal Value]	Velocity (m/s) [Normal Value]
Sensory				
Rt. median	Digit-2	3.15 [≤3.5]	26.5 [≥20]	
Lt. median	Digit-2	3.50 [≤3.5]	19.7 [≥20]	
Rt. ulnar	Digit-5	2.75 [≤3.2]	29.6 [≥17]	
Lt. ulnar	Digit-5	2.85 [≤3.2]	26.5 [≥17]	
Rt. sural	Lat. malleolus	2.55 [≤4.2]	11.7 [≥6]	
Lt. sural	Lat. malleolus	2.70 [≤4.2]	9.3 [≥6]	
Rt. sup. peroneal	Foot	3.05 [≤4.4]	9.3 [≥6]	
Lt. sup. peroneal	Foot	3.30 [≤4.4]	5.9 [≥6]	
Rt. MAC	Forearm	1.90 [≤3.2]	10.7 [≥5]	
Lt. MAC	Forearm	1.90 [≤3.2]	8.5 [≥5]	
Rt. LAC	Forearm	1.70 [≤3.0]	25.7 [≥10]	
Lt. LAC	Forearm	No response		
Motor				
Rt. median	APB	4.00 [≤4.1]	12.1 [≥5.5]	47.5 [≥50]
Lt. median	APB	4.15 [≤4.1]	9.8 [≥5.5]	44.7 [≥50]
Rt. ulnar	ADM	3.15 [≤3.1]	11.1 [≥5.9]	47.5 [≥52]
Lt. ulnar	ADM	3.05 [≤3.1]	10.7 [≥5.9]	45.8 [≥52]
Rt. common peroneal	EDB	5.25 [≤5.6]	4.2 [≥2.0]	40.6 [≥41]
Lt. common peroneal	EDB	5.35 [≤5.6]	3.8 [≥2.0]	37.2 [≥41]
Rt. tibial	AH	3.40 [≤5.4]	10.0 [≥4.6]	38.2 [≥42]
Lt. tibial	AH	3.55 [≤5.4]	9.9 [≥4.6]	39.0 [≥42]
Rt. axillary	Deltoid	2.25 [≤4.9]	12.3	
Lt. axillary	Deltoid	2.45 [≤4.9]	0.7	
Rt. musculocutaneous	Biceps	4.10 [≤5.7]	5.6	
Lt. musculocutaneous	Biceps	4.30 [≤5.7]	5.9	
Rt. suprascapular	SST	2.50 [≤3.7]	6.0	
Lt. suprascapular	SST	2.00 [≤3.7]	7.0	

MAC, medial antebrachial cutaneous nerve; LAC, lateral antebrachial cutaneous nerve; APB, abductor pollicis brevis; ADM, abductor digiti minimi; EDB, extensor digitorum brevis; AH, abductor hallucis; SST, supraspinatus.

## Data Availability

Not applicable.

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
