# Peer review of "Neuralgic Amyotrophy with Concomitant Hereditary Neuropathy with Liability to Pressure Palsy as a Cause of Dropped Shoulder in a Child after Human Papillomavirus Vaccination: A Case Report"

_children, 2022, doi:10.3390/children9040528_

Round 1
Reviewer 1 Report
Thank You for re-review of Your case report.
It is informative and interesting case and may be accepted in its present form.
All the best
Reviewer 2 Report
I express respect for the efforts of the authors who seriously responded to the strict questions from the reviewers. I think this paper has been improved to increase its scientific value.
Reviewer 3 Report
Thank you for giving me the opportunity to read and comment this case report.
It is a very interesting case.
It would be desirable for the authors to indicate the purpose of the highlighted parts of the manuscript.
It would be also interesting to mention previous medical history, previous vaccination and possible COVID-19 vaccination coverage.
Reviewer 4 Report
In this case report, the authors describe a case of neuralgic amyotrophy after human papillomavirus vaccination in a child patient with concomitant hereditary neuropathy with liability to pressure palsy. The case is interesting, and the authors provide details on the diagnostic procedures adopted for the final diagnosis. The results are clear overall, but I think it is important to clearly explain in figure 1 what each element of the figure represents (such as the box plots, the x-axis, the y-axis, the 3 colored areas, the red and blue lines, etc.) The discussion is good and includes appropriate data and references from the literature.
Author Response
Please see the attachment

This manuscript is a resubmission of an earlier submission. The following is a list of the peer review reports and author responses from that submission.
Round 1
Reviewer 1 Report
Thank for giving me the opportunity to revise this manuscript.
My impression is that is a well written case report that covers particular knowledge gaps of HPV vaccination and could be considered for publication in your prestigious journal after minor revision.
Specific Comments
Please add a paragraph in the discussion section describing in detail the advantages of HPV vaccination and review of the current evidence regarding the adverse effects. You can use as references the following articles:
- Sasieni P. Alternative analysis of the data from a HPV vaccine study in India. Lancet Oncol. 2022 Jan;23(1):e9. doi: 10.1016/S1470-2045(21)00661-6. PMID: 34973236.
- Huang R, Gan R, Zhang D, Xiao J. The comparative safety of human papillomavirus vaccines: A Bayesian network meta-analysis. J Med Virol. 2022 Feb;94(2):729-736. doi: 10.1002/jmv.27304. Epub 2021 Sep 4. PMID: 34453758
- Gidengil C, Goetz MB, Newberry S, Maglione M, Hall O, Larkin J, Motala A, Hempel S. Safety of vaccines used for routine immunization in the United States: An updated systematic review and meta-analysis. Vaccine. 2021 Jun 23;39(28):3696-3716. doi: 10.1016/j.vaccine.2021.03.079. Epub 2021 May 25. PMID: 34049735
Please also add a paragraph in discussion section describing the benefits as well as any potential effect of HPV vaccination on HPV tested positive women of reproductive age
You can use the following references
- Valasoulis G, Pouliakis A, Michail G, Kottaridi C, Spathis A, Kyrgiou M, Paraskevaidis E, Daponte A. Alterations of HPV-Related Biomarkers after Prophylactic HPV Vaccination. A Prospective Pilot Observational Study in Greek Women. Cancers (Basel). 2020 May 5;12(5):1164. doi: 10.3390/cancers12051164. PMID: 32380733; PMCID: PMC7281708
Reviewer 2 Report
Thank You very much for the opportunity to review this interesting case report.
It serves not only background for hypothesis of NA and HNPP but also bring pediatritians the knowledge about these disorders.
Minor comments:
- line 46: "Patiens with preexisting HNPP are vulnerable to the occurence of NA" - You cannot state this after just one case report. You can only hypothesize the link, but its relation may be only by chance not by relation. Please rewrite the sentence changing the meaning to "potential vulnerability"
- - line 50: "The study was approved.." and "Written informed consents.." should rather be in introduction/methods section. Its better to start describing the case by its real history ("12-year old female....") and not with methods. Please rewrite.
- - line 126/127 - it would be more informative to say few words about the presentation of cases in the existing literature. Additionally to discuss (if it is possible) potential differences betweent historical cases and the present case, in order to find potential factors that might suggest HNPP in patients with NA. This would be of merit for future clinical use.
